# Azomethine Ylides—Versatile Synthons for Pyrrolidinyl-Heterocyclic Compounds

**DOI:** 10.3390/molecules28020668

**Published:** 2023-01-09

**Authors:** Siva S. Panda, Marian N. Aziz, Jacek Stawinski, Adel S. Girgis

**Affiliations:** 1Department of Chemistry and Physics, Augusta University, Augusta, GA 30912, USA; 2Department of Pesticide Chemistry, National Research Centre, Dokki, Giza 12622, Egypt; 3Institute of Bioorganic Chemistry, Polish Academy of Sciences, Noskowskiego 12/14, 61-704 Poznan, Poland; 4Department of Organic Chemistry, Arrhenius Laboratory, Stockholm University, S-106 91 Stockholm, Sweden

**Keywords:** cycloaddition, azomethine ylide, pyrrolidine, spiro-compound

## Abstract

Azomethine ylides are nitrogen-based three-atom components commonly used in [3+2]-cycloaddition reactions with various unsaturated 2π-electron components. These reactions are highly regio- and stereoselective and have attracted the attention of organic chemists with respect to the construction of diverse heterocycles potentially bearing four new contiguous stereogenic centers. This review article complies the most important [3+2]-cycloaddition reactions of azomethine ylides with various olefinic, unsaturated 2π-electron components (acyclic, alicyclic, heterocyclic, and exocyclic ones) reported over the past two decades.

## 1. Introduction

The three-atom component (TAC) is an organic species that is represented by zwitterionic octet structures and undergoes [3+2]-cycloadditions with an unsaturated 2π-electron component in a one-step reaction, often in an asynchronous and symmetry-conducive fashion, via a thermal six-electron Hückel aromatic transition state. The formal charges are lost in the [3+2→5] cycloaddition (Figure 1) [1]. Recently, studies based on molecular electron density theory (MEDT) have suggested that the compounds involved in these reactions do not have a polar nature but a diradical, pseudoradical, or carbenoid nature. Therefore, the use of the term “1,3-dipole” is unjustified and should be replaced with “three-atom component”. It was also recommend that the designation of “dipolarophile” should be replaced with “unsaturated 2π-electron component”, and “1,3-dipolar cycloaddition” with “[3+2]-cycloaddition” [2].

While there is a mechanistic spectrum of this reaction from a synchronous one-step process to a stepwise overall transformation (including radical pathways), to avoid mechanistic digressions that may not have chemical or stereochemical consequences, in this synthetic review article, we will refer to the azomethine ylide reaction as a pericyclic cycloaddition. [3+2]-Cycloadditions of azomethine ylide with homomultiple and heteromultiple unsaturated 2π-electron components have been extensively used to produce a wide range of heterocycles [3]. There are several methods for the formation of azomethine ylides, including the thermolysis or photolysis of readily prepared aziridines, the dehydrohalogenation of immonium salts, and proton abstraction from imine derivatives of *α*-amino acids [3]. They are often generated in situ because of their high reactivity and/or transient existence; however, in some cases, stabilized ylides have been isolated and used further [4,5,6].

The synthesis of five-membered heterocyclic systems through azomethine ylides is one of the most adopted, efficient, and powerful approaches. Since the first report of successful the enantioselective [3+2]-cycloaddition of an azomethine ylide in 1991 [7], there has been tremendous progress in the chemistry regarding azomethine ylides. Azomethine ylides are extensively used in the synthesis of various heterocyclic systems such as pyrrolidines, pyrrolizidines, indolizidines, piperidines, oxazolidines, spiroindoles, spiropyrrolidines, and spiropiperidines, but they are also used for the total synthesis of complex natural products as well as bioactive compounds [8,9,10,11,12,13,14,15]. In recent years, the [3+2]-cycloaddition reaction has been extensively studied for the synthesis of heterocycles using different synthetic strategies [16,17]. In addition, the reaction is also investigated to understand the related reactivity, reaction conditions, intermediates, etc. [18,19].

This review article deals with the [3+2]-cycloaddition reaction of azomethine ylides with an unsaturated carbon–carbon bond (in either acyclic, alicyclic, heterocyclic, or exocyclic systems) that leads to the formation of pyrrolidinyl-containing analogs reported in the last two decades and their biological applications. This review article is intended to be a critical resource for the researchers involved or interested in azomethine ylides-mediated heterocyclic synthesis. It is also hoped that this review article will inspire chemists in this area of research.

## 2. Acyclic Unsaturated 2π-Electron Components

### 2.1. Intermolecular Cycloaddition Reaction of Azomethine Ylides to Acyclic Unsaturated 2π-Electron Components (Alkenes)

Unstabilized azomethine ylide **2** derived from benzyl(methoxymethyl)(trimethylsilylmethyl)amine **1** undergoes a [3+2]-cycloaddition reaction with electron-deficient alkenes **3** under continuous flow conditions in the presence of catalytic trifluoroacetic acid, thereby affording the corresponding pyrrolidines **4** (Figure 1) [20].

Azomethine ylides generated via the deprotonation of *α*-imino-esters **5** undergo a [3+2]-cycloaddition reaction with unsaturated 2π-electron components **6** in the presence of the eco-friendly supported solid-base catalyst KF/Al_2_O_3_ to yield the corresponding pyrrolidines **7** with high regio- and diastereoselectivity (Figure 2) [21].

Belfaitah et al. reported the cycloaddition reaction of azomethine ylides **9** with alkenyl boronates **8** to obtain the 3-boronic-ester-substituted pyrrolidines **10** (Figure 3) [22].

Pyrrolo[2,1-*a*]isoquinolines **15** were obtained through a sequential one-pot, two-step tandem reaction of isoquinoline **11**, *α*-halogenated methylenes **12**, aromatic aldehydes **13**, and cyanoacetoamide **14** in the presence of triethylamine as a basic catalyst and 2,4-dichloro-5,6-dicyano-1,4-benzoquinone (DDQ) as an oxidizing agent. The transformation was assumed to take place through [3+2]-cycloaddition of *N*-substituted carbonylmethyleneisoquinolinium bromide (formed via the reaction of isoquinoline **11** and **12**) with arylidene cyanoacetamide (formed via the condensation of cyanoacetamide **14** with aromatic aldehyde **13**) [23]. In the case of the ethyl bromoacetate **16** derivative, the formation of pyrrolo[2,1-*a*]isoquinolines **17** was observed probably due to DDQ oxidation (Figure 4) [23].

Spiro[indoline-3,2′-pyrrolidines] **21** were prepared by the [3+2]-cycloaddition reaction of benzoimidazol-2-yl-3-phenylacrylonitriles **18** with azomethine ylides, which was generated in situ from the condensation of isatin **19** and sarcosine **20** in refluxing ethanol. Similarly, spiro[indoline-3,5′-pyrrolo[1,2-*c*]thiazoles] **23** were formed by using thioproline **22** as a secondary amino acid (Figure 5) [24].

The chemistry was extended further to obtain spiro[acenaphthylene-1,2′-pyrrolidines] **26** and spiro[acenaphthylene-1,2′-pyrrolizidines] **28** possessing a cyano group from the azomethine ylides (generated from acenaphthenequinone **25**) with *α*-amino acids (sarcosine **20** and proline **27**) and Knoevenagel adducts **24** (Figure 6) [25].

### 2.2. Nitroalkenes

Nitroalkenes are reactive, unsaturated 2π-electron components that are intensively used in cycloaddition reactions by various researchers [26]. 3-Nitro-4-(trichloromethyl)pyrrolidine **30** was obtained through the cycloaddition of trans-3,3,3-trichloro-1-nitroprop-1-ene **29** with azomethine ylide (obtained from the condensation of paraformaldehyde and sarcosine in refluxing benzene). Quantum chemical calculations (DFT, M062X/6-311G(d)) explained the reaction pathway [27]. Analogously, 3-nitro-4-arylpyrrolidine-3-carbonitriles **32** were obtained through the cycloaddition of the azomethine ylide with (2*E*)-3-phenyl-2-nitroprop-2-enenitriles **31** [28] (Figure 7).

*Trans*-3-nitropyrrolidine **34** was prepared by reacting *trans*-1-nitro-2-phenylethylene **33** with *N*-(methoxymethyl)-*N*-[(trimethylsilyl)methyl]benzylamine **1**, which is an azomethine ylide equivalent, in the presence of trifluoroacetic acid in dichloromethane. Some of the synthesized **34** revealed promising inhibitory properties as Na^+^ channel blockers, which are useful in the treatment of ischemic stroke (Figure 8) [29].

Another set of spiro compounds, spiro[pyrrolidine-2,3′-oxindoles] **37,** were regioselectively synthesized by a multicomponent reaction of azomethine ylides, generated in situ from 3-aminoindoline-2-ones hydrochloride **35**, with aldehydes **13** and (*E*)-nitroalkenes **36** (Figure 9) [30].

It was assumed that, based on the secondary orbital interaction (SOI) of the electron-poor nitroalkenes **36** with the azomethine ylide, Path A was exclusively followed, as the *endo*-transition state in the reaction sequence was more energetically favorable (Figure 10) [30].

Spirooxindolo-nitropyrrolizines **38** (major product) and **39** (minor product) were obtained from the cycloaddition reaction of azomethine ylides, generated in situ from isatin **19**, with proline **27** and (*E*)-*ß*-nitrostyrene **32** (Figure 11) [31]. A significant inversion in the regioselectivity was observed when the polar [3+2]-cycloaddition of the azomethine ylides was attempted with trans-β-nitrostyrene instead of (*E*)-1-phenyl-2-nitropropene.

It was assumed that the reaction proceeds through *S*-shaped ylide with a cycloaddition via the endo-transition state (pathway B), yielding cycloadducts **38**, and not the exo-transition state (pathway A). Computational studies (Gaussian 03) of the transition states (Density Functional Theory (DFT), B3LYP, and 6-31G(d,p) basis set) confirmed these assumptions (Figure 12) [31].

A series of spiro[indoline-3,3′-pyrrolizin]-2-ones **40** with potential anti-amyloidogenic properties useful against Alzheimer’s disease were obtained by the microwave-assisted cycloaddition of nitroalkenes **36** and azomethine ylides (generated from isatin **19** and *L*-proline **27**) [32]. Analogously, spirooxindole-pyrrolidines **42** were obtained by the reaction of tyrosine **41** in an ionic liquid [bmim]Br at 100 °C. Promising antiproliferation properties were observed for some of the synthesized compounds (**42**) against human A549 (adenocarcinoma basal epithelial) and Jurkat (*T*-cell lymphoma) cell lines (MTT assay) using Camptothecin as a positive control; the compounds exhibited a safe response against the non-cancer cell lines MCF-10 (normal breast) and PCS-130-010 (lung smooth muscle). Caspase-dependent apoptosis (especially caspase-3) was mentioned as the mode of action for the observed antiproliferative activity (Figure 13) [33].

Ionic liquid chemistry was utilized to prepare 4′-nitrospiro[indeno[1,2-*b*]quinoxaline-11,2′-pyrrolidines] **47** by the cycloaddition reaction of nitroalkenes **36** with azomethine ylide (generated from indenoquinoxalinone **45** and *L*-phenylalanine **46**) in an ionic liquid [bmim]Br. Some of the synthesized agents revealed antimycobacterial properties (*Mycobacterium tuberculosis* H37Rv) with an efficacy comparable to that of ethambutol (reference standard) [34]. Similarly, spiro compounds **49** were obtained by using *L*-histidine **48** instead of *L*-phenylalanine **46** in this reaction. Some of the synthesized compounds revealed cholinesterase (acetylcholinesterase and butyrylcholinesterase)-inhibitory properties with considerable efficiencies relative to Galantamine (Figure 14) [35].

Pyrrolidinyl *ß*-lactams **52** were prepared as single diastereomers by the reaction of azomethine ylides **51**, generated from β-lactam imines of α-amino ester **50**, with nitrostyrenes **36** in the presence of silver acetate and triethylamine (Figure 15). This reaction is an example of [3+2]-cycloaddition reaction via *N*-metallo azomethine ylide [36].

3,4-Dihydropyrrolo[2,1-*a*]isoquinolines **54** were obtained by the [3+2]-cycloaddition reaction of nitroalkenes **36** with an azomethine ylide that was efficiently generated via the dirhodium(II)caprolactamate [Rh_2_(cap)_4_] catalyzed oxidation of tetrahydroisoquinoline **53** (Figure 16). Doyle’s oxidative protocol was used to generate azomethine ylides, which were further trapped in situ via [3+2]-cycloaddition [37].

### 2.3. α,β-Unsaturated Polarophiles

Spiro[3*H*-indole-3,3′-[3*H*]pyrrolizin]-2-ones **56** were synthesized by the cycloaddition reaction of (*E*)-3-aryl-1-(thiophen-2-yl)-prop-2-en-1-ones **55** with azomethine ylide generated in situ from the condensation of isatin **19** with *L*-proline **27** (Figure 17). Some of the synthesized spiroindoles **56** showed potential antibacterial activity against *Staphylococcus aureus* and *Salmonella typhi* (relative to Streptomycin) and antifungal activity against *Candida albicans* (relative to Amphotericin B) [38].

Spiro[pyrrolidine-2,3′-indolin]-2′-ones **59** were synthesized by the multi-component cycloaddition reaction of chalcones **58** and an azomethine ylide formed from the condensation of isatin **19** and benzylaminemine **57**. Few of the synthesized spiro-analogs **59** revealed potent inhibitory advanced glycation end (AGE) product formation in a bovine serum albumin (BSA)-glucose assay that was higher than that of aminoguanidine (standard reference). The occurrence of AGE is related to hyperglycemia observed as a complication of diabetes (Figure 18) [39].

Taghizadeh et al. reported an efficient and greener multicomponent protocol for the synthesis of regio-, diastereo-, and enantioselective spiro-oxindolopyrrolizidines **61** from optically active cinnamoyl oxazolidinone **60** and azomethine ylides that were formed from the condensation reaction of isatin **19** and *S*-proline **27** (Figure 19) [40].

Spiro[indoline-3,2′-pyrrolidines] **63** were prepared by the reaction of compound **62** containing an *α*,*β*-unsaturated ketone function with azomethine ylides obtained from isatin **19** and sarcosine **20**, while spiro[indoline-3,5′-pyrrolo[1,2-*c*]thiazoles] **64** was obtained from a similar reaction that involved thioproline **22** instead of sarcosine **20** (Figure 20). Some of the synthesized spiro-compounds, **63** and **64,** revealed anticancer properties against the A549 lung cancer cell line (MTT assay) [41,42] and spiro-compound **63** also showed antimicrobial activity against Gram-positive (*Micrococcus luteus*, *Enterobacter aerogenes*, *Staphylococcus aureus* and *Staphylococcus aureus* “MRSA-methicillin resistant”) and Gram-negative (*Salmonella typhimurium*, *Klebsiella pneumoniae*, *Proteus vulgaris*, and *Shigella flexneri*) bacterial strains and fungi (*Malassesia pachydermatis*, *Candida albicans*) relative to Streptomycin and Ketoconazole (used as antibacterial and antifungal standard references, respectively) [42].

Spiropyrrolidine-oxindoles **66** were prepared in appreciable yields by the cycloaddition reaction of the unsaturated 2π-electron component (*E*)-2-(1*H*-indole-3-carbonyl)-3-phenylacrylonitrile **65** and azomethine ylides obtained from the condensation of isatin **19** and sarcosine **20** (Figure 21) [43].

Similarly, spiropyrrolidine–oxindoles **68**–**70** were obtained from the reaction of enone **67** with azomethine ylides derived from isatin **19** and *α*-amino acids (sarcosine **20**, proline **27** or thioproline **22**). Among all the synthesized compounds, some showed antimicrobial properties against Gram-positive and Gram-negative bacterial as well as fungal strains using Streptomycin and Ketconazole as standard references (Figure 22) [44].

The unsaturated 2π-electron component, 2-[hydroxyl(4-oxo-4*H*-chromen-3-yl)methyl]acrylonitrile **71,** was synthesized by the Baylis–Hillman reaction of chromene-3-aldehyde, treated with the azomethine ylides (from isatin **19** and sarcosine **20**), which afforded the corresponding regioselective spiro[pyrrolidine-oxindoles] **72** and **73** as major and minor products, respectively (Figure 23) [45].

A convenient method for the selective construction of spiroindane-1,3-diones **77** relies upon the generation of unstabilized azomethine ylides from the initial condensation between ninhydrin **44** and 1,2,3,4-tetrahydroisoquinoline **74.** Subsequent azomethine ylide cycloaddition onto the conjugated double bond of chalcone **76** was exploited, giving target cycloadducts with good yields (77–94%) and diastereoselectivity (Figure 24) [46].

The reaction of azomethine ylide generated from 5-choloroisatin **19** and *L*-proline **27** as well as 1-acryloyl-4-piperidinones **78** yielded the corresponding spirooxindole-pyrrolizines **79** (yield 62–84%). Some of the synthesized cycloadducts **79** displayed cholinesterase-inhibitory properties (acetylcholinesterase and butyrylcholinestrase) with potency relative to Galantamine [47]. When the reaction was conducted in a 1:2:2 molar ratio of 1-acryloyl-4-piperidinones **78,** isatin **19,** and *L*-proline **27,** respectively, the bisspiropyrrolizines **80** were formed instead (yield 53–74%). It was found that most of the mono-spiropyrrolizines **79** (obtained using a 1:1:1 molar ratio of the reactants in yields of 73–84%) revealed higher cholinesterase enzyme (acetylcholinesterase and butyrylcholinestrase)-inhibitory activity than the bisspiropyrrolizine derivatives **80** (Figure 25) [48].

The reaction of 3-(3-phenylazetidin-2-yl) acrylates **81** with azomethine ylide formed by the condensation of ninhydrin **44** and amino acids (sarcosine **20**/*L*-proline **27**) afforded the corresponding spiroindanopyrrolidines **82** and spiroindanopyrrolizines **83** (Figure 26). The synthesized cycloadducts **82** and **83** showed antibacterial properties against *Proteus mirabilis*, *Proteus vulgaris*, *Salmonella typhi*, and *Staphylococcusi aureus* relative to Tetracycline (standard reference drug) [49].

Cycloaddition of cinnamaldehydes **84** with azomethine ylides, generated from another cinnamaldehyde molecule **84** and *L*-proline **27**, afforded hexahydro-1*H*-pyrrolizines **85** and **86** in different ratios depending on the heating method (conventional heating, 25–80 °C vs. with microwave technique) and the solvent used (MeCN, DMF, toluene, CH_2_Cl_2_, DMSO) (Figure 27) [50].

Pyrrolizidines of type **88** were obtained by reacting *β*,*γ*-unsaturated *α*-keto esters of type **87** with proline **27** in a 2:1 molar ratio. The reaction was assumed to proceed via the formation of azomethine ylides by the condensation of the starting unsaturated esters of type **87** with amino acid **27**, which, in turn, interacted with another molecule of **87** to ultimately yield pyrrolizidines of type **88** (Figure 28) [51].

### 2.4. Acrylates

The reaction of *O*-acryloylacridinediones **89** with azomethine ylides, generated from isatin **19** and secondary amino acids (sarcosine **20**/proline **27**), afforded the corresponding spiro-pyrrolidines **90** and spiro-pyrrolizidines **91** (Figure 29) [52].

Spiropyrrolidines **94**–**97** were obtained via the reaction of methyl 2-(1*H*-inden-2-yl)acrylate **92** with azomethine ylides generated in situ by reacting ketones (isatin **19**, acenaphthenequinone **25**, ninhydrin **44,** or 11*H*-indeno[1,2-*b*]quinoxaline-11-one **93**) with sarcosine **20** (Figure 30) [53].

The reaction of methyl lactate acrylates of type **98** with azomethine ylides, generated from imino-esters **5** in the presence of silver acetate and KOH, gave chiral proline derivatives of type **99** (Figure 31) [54].

The reaction of *trans* arylacrylates **100** with the azomethine ylide, formed from benzyl-(methoxymethyl)[(trimethylsilyl)methyl]amine **1** in the presence of a catalytic amount of trifluoroacetic acid, afforded the corresponding *trans* pyrrolidine derivatives **101** (Figure 32) [55].

### 2.5. Intramolecular Cycloaddition Reaction of Azomethine Ylides with Acyclic Unsaturated 2π-Electron Components

#### 2.5.1. Acyclicunsaturated 2π-Electron Components Containing Olefinic and Aldehyde Groups

Azomethine ylides (formed via the reaction of *α*-amino esters **103** with *O*-allyl-5-phenyldiazenylsalicylaldehyde **102**) underwent intramolecular [3+2]-cycloaddition under microwave conditions, affording the 8-phenyldiazenylchromeno[4,3-*b*]pyrrolidines **104** (Figure 33). The synthesized compounds showed antibacterial activity against Gram-positive (*Streptococcus pneumoniae*, *Clostridium tetani*, and *Bacillus subtilis*) and Gram-negative bacteria (*Salmonella typhi*, *Vibrio cholerae*, and *Escherichia coli*), fungi (*Aspergillus fumigatus* and *Candida albicans*), and mycobacteria (*M. Tuberculosis* H37RV) relative to the antibacterial (Ampicillin, Norfloxacin, Chloramphenicol, Ciprofloxacin), antifungal (Griseofulvin, Nystatin), and antimycobacterial (Metronidazole) standard references used [56].

The intramolecular cycloaddition reaction of azomethine ylides, formed from alkenyl aldehyde **105** and secondary amino acids (sarcosine **20**, *L*-proline **27**, thioproline **22,** and tetrahydroisoquinoline-3-carboxylic acid **106**), afforded the corresponding chromenopyrrole derivatives **107**–**109** (Figure 34). The synthesized compounds showed promising antibacterial (against *S. aureus*, *B. subtilis* “Gram-positive”; *S. pneumoniae*, *E. coli*, and *Shigella* sp., *S. typhi* “Gram-negative”) and antifungal (against *Trichoderma* sp., *Aspergillus* sp. and *C. albicans*) activities against the references Tetracycline and Carbendazim (antibacterial and antifungal standard references, respectively) [57].

The intramolecular cycloaddition of *O*-allyl salicylaldehydes **110** and sarcosine **20** under ultrasonic irradiation in methanol at room temperature yielded the corresponding chromeno[4,3-*b*]pyrroles **111** (Figure 35) [58].

Chromeno[4,3-*b*]pyrrolidines **113** were obtained in a highly regio- and stereoselective manner by the intramolecular cycloaddition of *O*-allylic salicylaldehydes **112** and sarcosine **20** (Figure 36) [59].

Similarly, hexahydrochromeno[4,3-*b*]pyrroles **116** were obtained via intramolecular [3+2}-cycloaddition of *O*-allylic salicylaldehyde **114** and amines **115** under microwave conditions (Figure 37) [60].

Bicyclic pyrrolo[3,4-*b*]pyrroles **118** were obtained by the intramolecular cyclization of the generated azomethine ylides from aldehydes **117** and sarcosine **20** under refluxing conditions in toluene (Figure 38) [61].

Octahydropyrrolo[3,4-*b*]pyrroles **121** with various substituents in their aromatic rings were synthesized by the intramolecular cycloaddition of azomethine ylides, which was formed from the reaction of alkenyl aldehyde **119** with *N*-aryl glycines **120** (Figure 39) [62].

The condensation of *N*-alkenyl aldehydes **122** with *α*-amino acids (sarcosine **20**, thioproline **22** and proline **27**) generated azomethine ylides, which underwent an intramolecular cycloaddition reaction yielding the corresponding polycyclic compounds **123** and **124** (Figure 40) [63].

Similarly, the intramolecular reaction of azomethine ylide obtained from 2-butenylindole-3-carboxaldehyde **125** with *N*-methyl glycine ethyl ester hydrochloride **126** gave the indole-containing alkaloid **127**. Whereas its reaction with *N*-methyl glycine **20** or *N*-allyl glycine **128** gave the corresponding indole heterocycles of type **129** (Figure 41) [64].

Another example of intramolecular cycloaddition was the reaction of (*E*)-2-{[allyl(benzyl)amino]methyl}cinnamaldehydes **130** with proline methyl ester hydrochloride **131** under microwave conditions, which afforded the pyrido[3,4-*b*]pyrrolizines **132** (Figure 42) [65].

By using 1,2-*O*-cyclohexylidine-3-*O*-allyl-*α*-*D*-xylopentadialdo-1,4-furanose **133** (sugar-derived aldehyde) in a reaction with sarcosine **20,** furopyranopyrrolidine of type **134** was formed with high diastereoselectivity (Figure 43) [66].

The intramolecular [3+‏2]-cycloaddition of azomethine ylides, generated from 2-formylphenyl-(*E*)-2-phenylethenesulfonates **135** and sarcosine **20,** afforded the corresponding [1,2]oxathiino[4,3-*b*]pyrroles **136**. However, the reaction of derivative **135** with *L*-proline **27** gave the corresponding [1,2]oxathiino[3,4-*b*]pyrrolizines **137** as *trans*–*trans* (major) and *cis*–*trans* (minor) isomers (Figure 44) [67].

Figure 45 shows an interesting example of a macrocycle of type **139** formation via the intramolecular cycloaddition of an azomethine ylide generated from a triazole-linked glycol-nitroalkenyl aldehyde derivative **138** and sarcosine **20** [68].

Polycyclic naphtho[2,1-*b*]pyrano-pyrrolizidine and indolizidine derivatives **141** and **143** were synthesized by the intramolecular [3+2]-cycloaddition of azomethine ylides generated from naphtho-*O*-alkenyl aldehydes **140** and *α*-amino acids (*L*-proline **27** or *DL*-pipecolinic acid **142)** (Figure 46) [69].

#### 2.5.2. Acyclic Unsaturated 2π-Electron Components Containing Olefinic Linkage and Azirdine

Figure 47 shows the thermolysis of aziridines **144** that led to the in situ formation of azomethine ylides, which underwent intramolecular cycloaddition, thus affording *N*-phthalimidopyrrolidine derivatives **145** as a mixture of two diastereoisomers [70].

Another bicyclic system of *γ*-lactone **147** was created by the intramolecular [3+2]-cycloaddition of azomethine ylide generated via the thermolysis of aziridine derivative **146** in refluxing toluene (Figure 48) [71].

## 3. Exocyclic, Unsaturated 2π-Electron Components

### 3.1. Cycloalkanones

The exocyclic olefinic linkage is a reactive, unsaturated 2π-electron component intensively used in [3+2]-cycloaddition reactions forming various heterocycles [72,73,74,75,76]. For example, the cycloaddition of azomethine ylide (formed from isatin **19** and sarcosine **20**) with 2-arylidene-1-cyclopentanones **148** in the presence of bentonite clay under microwave conditions afforded dispiropyrrolidinyl-oxindoles **149** (Figure 49) [77].

Similarly, dispiro[cyclohexane-1,3′-pyrrolidine-2′,3″-[3*H*]indoles] **151** and **152** were obtained by the cycloaddition reaction of azomethine ylides (generated from isatin derivative **19** and sarcosine **20**) with 2*E*,6*E*-bis(arylidene)-1-cyclohexanones **150** (Figure 50). Some of the synthesized compounds demonstrated antitumor properties against liver (HEPG2), cervical (HELA), and prostate (PC3) cancer cell lines while using Doxorubicin as a standard reference in an SRB assay [78].

Azomethine ylide formed from the condensation of benzylamine **57** and isatin **19** also underwent a cycloaddition reaction with 2,6-bis(ylidene)cyclohexanones **150** under solvent-free conditions using microwave irradiation, thereby affording the dispiro-oxindole **153** with high regioselectivity (Figure 51) [79].

Azomethine ylides derived from acenaphthenequinone **25** and *α*-amino acids (sarcosine **20**, phenylglycine **154**, proline **27,** or thioproline **22**) afforded the corresponding spiro-cyclohexanones **155**–**158** upon reaction with 2,6-bis(ylidene)cyclohexanones **150** in refluxing methanol [80] (Figure 52). Some of the synthesized spiro compounds revealed activity against *Mycobacterium tuberculosis* H37Rv (MTB) relative to Ethambutol and Pyrazinamide [80].

A one-pot, five-component reaction of azomethine ylide (formed from ninhydrin **44**, *o*-phenylenediammine **43**, and sarcosine **20**) with bis(ylidene)cycloalkanones **159** in the presence of hydrazine hydrate **160** in refluxing methanol regioselectively afforded the corresponding spiro-indenoquinoxaline-pyrrolidines **161** at a high yield (Figure 53) [81].

Trispiropyrrolidines/thiapyrrolizidines **163** and **164** were synthesized through the reaction of 7,9-bis[(*E*)-ylidene]-1,4-dioxa-spiro[4,5]decane-8-ones **162** and azomethine ylides (formed from isatin **19** and sarcosine **20** or thioproline **22**) in 2,2,2-trifluoroethanol (TFE) (Figure 54). Some of the products showed anti-fungal properties (against *Candida albicans* MTCC 227, *Aspergillus niger* MTCC 282, and *Aspergillus clavatus* MTCC 1323) and antimycobacterial properties against *M. tuberculosis* H37Rv relative to the standard references Nysyatin, Greseofulvin (antifungal), and Isoniazid (antimycobacterial) [82].

Analogously, dispiro compounds of type **165** were synthesized by the reaction of 2,6-bis(ylidene)cyclohexanones **150** with azomethine ylide (formed from *L*-thioproline **22** and isatin **19**) in refluxing methanol. Some of the synthesized derivatives revealed promising antiproliferative properties (apoptotic mechanism) against the MCF7 (breast) and K562 (leukemia) cell lines (WST-1 assay) relative to 5-Fluorouracil (standard reference drug) (Figure 55) [83].

### 3.2. Indanones and Indanediones

A series of dispiro compounds of type **167** were regioselectively synthesized by the cycloaddition of 2-(ylidene)-1-indanones **166** with azomethine ylides (formed from isatin derivatives **19** with sarcosine **20**) in refluxing ethanol. Promising anti-inflammatory properties were exhibited by the synthesized compounds (via a rat carrageenan paw edema assay) relative to Indomethacin (standard reference drug) [84]. Antiproliferative properties were also revealed by some of the synthesized derivatives against human metastatic melanoma cells (GaLa, LuPiCi, and LuCa), with a potency relative to that of Doxorubicin (SRB assay) (Figure 56) [85]. In an analogous reaction, by using *L*-thioproline **22** instead of sarcosine **20,** spiro-pyrrolothiazolyloxindole derivatives of type **168** were obtained. Some of these compounds showed activities against *Mycobacterium tuberculosis* H37Rv relative to Ethambutol (standard reference) [86].

Other dispiropyrrolidines of type **169** were synthesized by the cycloaddition of azomethine ylide (formed from ninhydrin **44** and sarcosine **20)** with 2-(arylidene)-1-indanones **166** (Figure 57). When acenaphthenequinone **25** was used instead of ninhydrin **44** in this reaction, dispiropyrrolidines of type **170** were formed in a highly regio- and stereoselective manner. Some of the synthesized derivatives—**169** and **170** showed antimycobacterial properties against *M. tuberculosis* H37Rv and INH resistant *M. tuberculosis* strains relative to Isoniazid and Ethambutol (standard reference drugs) [87,88].

Dispiropyrrolidines **171** and **172** were obtained by the cycloaddition of 2-(ylidene)-1-indanones **166** with azomethine ylides (obtained through the condensation of *L*-thioproline **22** with ninhydrin/acenaphthenequinone **44/25**) in refluxing methanol. Some of the synthesized compounds showed promising in-vitro antimycobacterial properties against *M. tuberculosis* H37RV relative to Cycloserine [89]. Analogously, pyrrolothiazolyloxindoles of type **173** were obtained when isatin **19** was used instead of ninhydrin **44** or acenaphthenequinone **25** in this reaction. Some of the isatin-derived compounds of type **173** exhibited inhibitory properties toward acetylcholinesterase that could be useful for Alzheimer’s disease therapy (Figure 58) [90].

By reacting 5,6-dimethoxy-2-(arylidene)-1-indanone **174** and isatin **19** with sarcosine **20** or phenylglycine **154**, spiropyrrolidines **175** and **176**, respectively, were obtained (Figure 59). Some of these compounds showed inhibitory activities toward acetylcholinesterase [91].

TiO_2_–silica was used as an efficient solid-supported catalyst for the cycloaddition reaction of 2-arylidene-1,3-indanediones **177** with the corresponding azomethine ylides generated from tetrahydroisoquinoline-3-carboxylic acid **106** and isatin derivative **19** or acenaphthenequinone **25** to afford the corresponding dispiropyrroloisoquinolines **178** and **179** (Figure 60) [92].

The four-component reaction of 2-arylidene-1,3-indanediones **177**, ninhydrin **44**, *o*-phenylenediamine **43**, and *L*-proline **27**, proceeding via an azomethine intermediate and in the presence of heteropolyacid H_4_[Si(W_3_O_10_)_3_]–silica as a catalyst in refluxing acetonitrile, afforded the dispiroindenoquinoxaline-pyrrolizidines **180** (Figure 61) [93].

Dispiro compounds of type **182** were synthesized by the reaction of the generated azomethine ylides (from isatin **19** and sarcosine **20**) with 2-(1,3-dioxo-indan-2-ylidene)malononitrile **181** (Figure 62) [43].

### 3.3. Fluorenes

The solvent-free reaction of (*E*)-arylidenefluorenes **183** with isatin **19** and sarcosine **20** or proline **27,** under microwave conditions, afforded the corresponding dispiro-oxindoles **184** and **185**, respectively (Figure 63) [94].

### 3.4. Acenaphthenes

The reaction of acenaphthenone-2-ylidene ketones of type **186** with azomethine ylides formed from the condensation of isatin **19** or acenaphthenequinone **25** and secondary amino acids (sarcosine **20** or *L*-proline **27**) in refluxing methanol afforded the corresponding spirooxindoles **187**–**190** (Figure 64) [95].

Similarly, 2-oxo-(2*H*)-acenaphthylen-1-ylidene-malononitrile **191** afforded the corresponding dispiropyrrolidine-oxindoles **192** by its reaction with isatin **19** and sarcosine **20** in refluxing toluene (Figure 65) [96].

### 3.5. Tetralones

Dispiro-oxindolopyrrolidine/pyrrolizidines **194** and **195** were synthesized via the cycloaddition of (*E*)-1-naphthylidene-1-tetralone **193** with the corresponding azomethine ylides generated from isatin **19** and sarcosine **20** or *L*-proline **27** (Figure 66) [97].

Further, the cycloaddition of 1,4-bis(3′,4′-dihydro-1′-oxonaphthalen-2′-ylidene)benzene derivative **196** with azomethine ylides (from isatin **19** and sarcosine **20**) in a 1:2 molar ratio afforded the corresponding tetraspiro-bisoxindolopyrrolidine **198**. With a 1:1 ratio of the reactants, mono derivatives of type **197** were formed, which, in the presence of an excess of isatin **19** and sarcosine **20,** afforded bisoxindolopyrrolidines **198** [98] (Figure 67).

### 3.6. Pyrrolidine-2,5-diones

Dispiropyrrolidines of type **200** were prepared regioselectively by the cycloaddition of 3-(ylidene)pyrrolidine-2,5-diones **199** with azomethine ylide (formed from condensation of sarcosine **20** and isatin **19**) in refluxing alcohol. Promising cholinesterase (acetylcholinesterase and butyrylcholinesterase) inhibitory properties were observed for some of the synthesized compounds (relative to Donepezil, used as the standard reference) that are of potential importance for fighting Alzheimer’s disease (Figure 68) [99]. Some of the synthesized **200** also revealed antibacterial activities against *Bacillus subtilis* NCIM 2718, *Staphylococcus aureus* NCIM5021, *Salmonella typhi* NCIM2501, *Pseudomonas aeruginosa* NCIM 5029, and *Proteus vulgaris* NCIM2813 relative to Ampicillin [100].

### 3.7. Lactones

Dispiropyrrolidino/pyrrolizidino-oxindoles **202** and **203** were obtained through the cycloaddition of *α*,*ß*-unsaturated-*γ*-lactone **201** with isatin **19**/sarcosine **20** or isatin **19**/proline **27** reagent systems (Figure 69) [101].

Glycospiro-2,3-dihydropyrrolo[2,1-*a*]isoquinolines **206** were synthesized by the reaction of 3-deoxy-3-*C*-[(*Z*)-(methoxycarbonyl)methylene]-1,2:5,6-di-*O*-isopropylidene-*α*-*D*-glucofuranose **205** with isoquinoline-based azomethine ylide formed from isoquinolines **11** and alkyl bromoacetates or 2-bromoacetophenones **204** in the presence of Cu(OTf)_2_–Et_3_N as a catalyst (Figure 70) [102].

### 3.8. Thiophenones

The reaction of 2-(ylidene)thiophen-3-ones **207** with various azomethine ylides generated from sarcosine **20** and different ketones (isatin **19**, ninhydrin **44** or acenaphthoquinone **25**) afforded the corresponding dispiropyrrolidine containing-thiophenones **208**–**210** in good yields (Figure 71) [103].

### 3.9. Oxazolones

The reaction of 4-ylidene-5-(4*H*)-oxazolones **211** with azomethine ylide derived from *cis*-4-formyl-2-azetidinone **212** and sarcosine **20** or pyrrolidine **27**, in the presence of camphor sulphonic acid (CSA) as a catalyst, afforded the corresponding spiro[3.4′]-(oxazol-5′-one)-pyrrolidines **213** and spiro[3.4′]-(oxazol-5′-one)-pyrrolizidines **214**, respectively (Figure 72) [104].

Spiro-compounds of types **216**–**219** were obtained via the cycloaddition reaction of 4-ylidene-5-oxazolones **211** and azomethine ylides (generated from isatin **19** and the appropriate *α*-amino acids). Some of the synthesized compounds showed considerable antitumor properties against breast cancer (MCF7, MDA-MB-231, and MDA-MB-468) and hepatocellular (HepG2, HCCC-9810, and HuH7) cell lines (MTT assay) relative to Gefitinib and Sorafenib (standard references) (Figure 73) [105].

### 3.10. Indoles

The azomethine ylide (formed from isatin **19** and sarcosine **20**) underwent cycloaddition with 3-aroylmethyleneindol-2-ones **220** under green chemistry conditions in an ionic liquid ([bmim]PF_6,_ 1-butyl-3-methylimidazolium hexafluorophosphate) to afford the corresponding dispiropyrrolidine-bisoxindoles **221** [106]. TiO_2_–silica was also used as a solid-supported catalyst under microwave conditions for the synthesis of dispiropyrroloisoquinolines **222** via the three-atom component cycloaddition reaction of azomethine ylide (generated from tetrahydroisoquinoline-3-carboxylic acid **106** and isatin **19**) with **220** [92] (Figure 74).

Dispiro-oxindolopyrrolizidines **223** and dispiro-oxindolothienopyrroles **224** could also be obtained by the reaction of azomethine ylides (generated from isatin **19** with *L*-proline **27** or *R*-thioproline **22**) with 3-aroylmethyleneindol-2-ones **220** under ultrasonication conditions in the presence of silica as a catalyst (Figure 75) [107].

Similarly, the cycloaddition reaction of 3-aroylmethyleneindol-2-ones **220** with azomethine ylide (formed from tetrahydroisoquinoline-3-carboxylic acid **106** and acenaphthenequinone **25**) using TiO_2_–silica as a solid-supported catalyst under microwave conditions yielded dispiropyrroloisoquinoline **225** [92]. Ball-clay-supported zirconium oxychloride octahydrate was also used as a catalyst in the cycloaddition reaction of azomethine ylides (generated from acenaphthenequinone **25** and sarcosine **20** or *L*-proline **27**) to produce spiro-oxindolopyrrolidine **226** and spiro-oxindolopyrrolizidine **227**, respectively (Figure 76) [108].

Spiro-oxindoles of type **229** were obtained by reacting isatin **19**, sarcosine **20**, and 3-[2-oxo-ethylidene]indolin-2-one **228** in equimolar quantities whereas spiro-oxindole derivatives of type **230** were formed when isatin **19** and sarcosine **20** were used at a two-fold degree of molar excess over acceptor **228** (Figure 77) [109].

Another dispirocyclopentanebisoxindole **232** can be obtained by the cycloaddition of azomethine ylides generated from 4-dimethylamino-1-alkoxycarbonylmethylpyridinium bromide **231** and aroylmethyleneindol-2-one **220** (Figure 78) [110].

The cycloaddition of aroylmethyleneindol-2-one **220** with azomethine ylide formed from indenoquinoxaline-11-one (generated in situ from ninhydrin **44** and *o*-phenylenediamine **43**) and *L*-proline **27** was catalyzed by eteropolyacid H_4_[Si(W_3_O_10_)_3_]–silica and afforded the corresponding dispiroindenoquinoxaline-pyrrolizidine **233** (Figure 79) [91].

The reaction of azomethine ylides—formed from the condensation of isatin **19** and sarcosine **20** or proline **27** in an ionic liquid [bmim] BF_4_ (without using any catalyst)—with 2-cyano-2-(2-oxoindolin-3-ylidene)acetate **234** yielded the corresponding dispiropyrrolidine-bisoxindole **235** and dispiropyrrolizidine-bisoxindole **236**, respectively (Figure 80) [111].

Spiropyrrolidine/spiropyrrolizine-oxindoles **238**–**241** were synthesized by a multi-component cycloaddition reaction of 2-oxo-(3*H*)-indol-3-ylidine-malononitrile **237** with azomethine ylides (generated from aromatic aldehyde **13** and sarcosine **20** or *L*-proline **27**) in refluxing toluene containing molecular sieves (3 Å) (Figure 81) [112].

In addition, dispiropyrrolidine-bisoxindole derivatives of type **242** were obtained from a three-atom component reaction of isatin **19**, sarcosine **20,** and isatylidene malononitrile **237** with high regioselectivity (Figure 82) [43].

Dispiro-oxindoles of type **243** were obtained by the dimerization of in situ generated azomethine ylides (A and B) via A+B pathways. X-Ray studies supported the postulated structures of type **243** (Figure 83) [113].

### 3.11. Benzofuran-2-ones

Dispiro-oxindolopyrrolidine **245** and dispiro-oxindolopyrrolothiazole **246** were obtained by a multi-component reaction of 3-(ylidene)benzofuran-2-one **244** with isatin **19** and the appropriate *α*-amino acid (sarcosine **20** or thioproline **22**, respectively) in refluxing methanol. Some of the synthesized compounds revealed promising antimycobacterial (*M. tuberculosis* H37Rv) properties relative to pyrazinamide (standard reference) (Figure 84) [114].

### 3.12. Keto-Carbazoles

Dispiro[carbazole-2,3′-pyrrolo-2′,3″-indole] derivatives of type **248** were synthesized regio- and stereoselectively by the reaction of 2-ylidene-1*H*-carbazol-1-one **247** and in situ-generated azomethine ylide (formed from isatin **19** and benzylamine **57**). Some of the obtained compounds revealed antiproliferative properties (MTT assay) via apoptosis induction against MCF7 (breast) and A-549 (lung) cancer cell lines relative to Cisplatin (Figure 85) [115].

The reaction of (*E*)-2-arylidine-1-ketocarbazole **247** with various azomethine ylides generated from sarcosine **20** and di/tri ketone (isatin **19**, ninhydrin **44**, acenathenequinone **25**) under microwave irradiation afforded the corresponding ketocarbazolodispiropyrrolidines **249**–**251** (Figure 86). Some of the synthesized compounds showed antimicrobial properties against *Proteus vulgaris*, *Proteus mirabilis*, *Staphylococcus aureus,* and *Salmonella typhi* relative to Tetracycline [116].

Similarly, dispiro-oxindolopyrrolizidines of type **252** were prepared by reacting (*E*)-2-arylidine-1-ketocarbazole **247** with the azomethine ylide formed from isatin **19** and proline **27** (Figure 87). Some of the products showed antimicrobial activities against human pathogens (*Proteus vulgaris*, *Proteus mirabilis*, *Staphylococcus aureus*, and *Salamonella typhi*) relative to Tetracycline and acted as inhibitors of plant fungal pathogen mycelial growth (*Fusarium oxysporum* and *Macrophomena phaseolina*) relative to the standard reference—Carbendazim [117].

### 3.13. Piperidones

The reaction of isatin **19** with various amines (sarcosine **20**, proline **27,** and benzylamine **57**) in refluxing methanol or in ionic liquid [bmim]Br generated the corresponding azomethine ylides that was added to 3-(arylidene)-4-piperidones **253** to form the corresponding spiropiperido-pyrrolizines/pyrrolidines **254**–**256** (Figure 88). Some of the products showed activities against *Mycobacterium tuberculosis* H37Rv (MTB), multi-drug resistant *M. tuberculosis* (MDR-TB), and *Mycobacterium smegmatis* (MC2) relative to ethambutol and pyrazinamide (standard references) [118], and also had acetyl- and butyrylcholinesterase inhibitory properties (of potential use against Alzheimer’s disease) relative to galantamine [119].

A variety of dispiro-heterocycles of types **260**–**263** were obtained via azomethine ylide intermediates (formed from isatin **19** and sarcosine **20**, piperidine-2-carboxylic acid **258**, thioproline **22,** or 2-amino-3-phenylpropanoic acid **259**) in a reaction with 3,5-bis(ylidene)-4-piperidone **257** (Figure 89). Some of the synthesized analogs revealed considerable antiproliferation properties against a variety of tumor cell lines [120,121,122,123]. Promising anti-inflammatory properties were also exhibited by some of the synthesized compounds (50 mg/kg) in a rat model of carrageenan-induced paw edema (anti-edematous test) relative to indomethacin (10 mg/kg) [120,124]. Some derivatives of compound **262** showed activities against *Mycobacterium tuberculosis* H37Rv (MTB) and multi-drug resistant *M. tuberculosis* (MDR-TB) relative to ethambutol and pyrazinamide (standard references) [125], and had antifungal properties against *Candida albicans* ATCC 10231 with high inhibition of the fungal hyphae relative to fluconazole (standard reference drug) [126].

Similarly, the reaction of 3,5-bis(ylidene)-4-piperidone **257** with a series of azomethine ylides generated from acenaphthenequinone **25** and *α*-amino acids (sarcosine **20**, phenylglycine **154**, proline **27**, thioproline **22,** or piperidine-2-carboxylic acid **258**) afforded the corresponding spiropiperidone-containing compounds **264**–**268** (Figure 90). Some of these derivatives revealed promising activities against *Mycobacterium tuberculosis* H37Rv (MTB), multi-drug resistant *Mycobacterium tuberculosis* (MDR-TB), and *Mycobacterium smegmatis* relative to Isoniazid [127]. Another group of the synthesized spiro-heterocycles **267** and **268** showed acetylcholine (AChE)-inhibitory properties relative to Donepezil HCl [128].

A multicomponent reaction of 3,5-bis[(*E*)-ylidene]-4-piperidone **257**, ninhydrin **44**, *o*-phenylenediamine **43**, and *α*-amino acid (sarcosine **20** or *L*-tryptophan **269**) in 1-butyl-3-methylimidazoliumbromide ([BMIm]Br) used as an ionic liquid produced the corresponding dispiro compounds **270** and **271** [129,130] (Figure 91). Significant acetylcholinesterase- (AChE) and butyrylcholinesterase (BChE)-inhibitory properties were shown by some of the synthesized compounds relative to galantamine (standard reference) [130].

Mono-spiropyrrolidines of type **272** were synthesized by the reaction of 1-acryloyl-3,5-bis(ylidene)-4-piperidinone **78** with azomethine ylide generated from isatin **19** and phenylglycine **154** from equimolar amounts of the reactants. Meanwhile, bisspiropyrrolidine derivatives of type **273** were formed using two equivalents, namely, isatin **19** and phenylglycine **154** (Figure 92). Some of the synthesized compounds showed promising AChE- and BChE-inhibitory properties relative to Galanthamine [131].

Finally, the reaction of 3,5-bis(ylidene)-4-piperidone **257** with azomethine ylide formed from ninhydrin **44** and proline **27** in refluxing methanol afforded diazahexacycle **274** [132]. Similarly, **275**–**277** were obtained by the reaction of **257** with another azomethine ylides generated from ninhydrin **44** or acenaphthenequinone **25** with sarcosine **20** or *L*-phenylalanine **45** (Figure 93) [133,134]. Some of the derivatives of type **274** exhibited inhibitory activities toward AChE relative to Donepezil HCl [132].

### 3.14. Quinolones

Spiropyrrolidines **279** and **280** were synthesized by the reaction of (*E*)-3-(ylidene)-4-quinolone **278** with azomethine ylides formed from isatin **19** and sarcosine **20** or thioproline **22** (Figure 94) [135].

### 3.15. Chromanones

The reaction of (*E*)-3-arylidene-4-chromanone of type **281** with azomethine ylides formed from acenaphthenequinone **25** and sarcosine **20** or proline **27** afforded spiropyrrolidines **282** and **283**, respectively, with high regioselectivity (Figure 95) [136].

A multi-component reaction of 2-ylidene-tetrahydronaphthalene-1-one **284** or (*E*)-3-ylidene-4-chromanone **281** with azomethine ylide formed from indenoquinoxaline-11-one (generated from ninhydrin **44** and *o*-phenylenediamine **43**) and *L*-proline **27** afforded, in the presence of heteropolyacid H_4_[Si(W_3_O_10_)_3_]‒silica as a catalyst in refluxing acetonitrile, the corresponding dispiroindenoquinoxaline-pyrrolizidines **285** and **286**, respectively (Figure 96) [93].

### 3.16. Thiochromanones

Dispiro[indene-2,2′-pyrrolidine-3′,3″-thiochromanes] **288** were obtained by reacting 3-arylidenethiochroman-4-one **287** with azomethine ylide (obtained from ninhydrin **44** and sarcosine **20**) in refluxing methanol. When thioproline **22** was used instead of sarcosine **20**, the corresponding dispiro derivatives of type **289** were obtained. Some of the synthesized pyrrolizidines revealed antimycobacterial properties (*Mycobacterium tuberculosis* H37Rv) relative to Cycloserine and Pyrimethamine (standard references). Additionally, mild antiproliferative properties against CCRF-CEM (leukemia), HT29 (ovarian), and MCF7 (breast) cancer cell lines relative to Doxorubicin (MTT assay) were observed (Figure 97) [137].

### 3.17. Acridinones

The reaction of (*E*)-2-(arylidene)-3,4-dihydro-1(2*H*)-acridinones **290** with azomethine ylides formed from the condensation of isatin **19** and sarcosine **20** or thioproline **22** in refluxing dioxane/methanol afforded the corresponding dispirooxindolyl-[acridine-2′,3-pyrrolidine/thiapyrrolizidine]-1′-ones **291** and **292**, respectively (Figure 98) [138].

Naphtho[1′,8′:1,2,3]pyrrolo[3′,2′:8,8*a*]azuleno[5,6-*b*]quinolin-14-one **293** and naphtho[1′,8′:1,2,3]thiazolo[3″,4″:1′,5′]pyrrolo[3′,2′:8]-azuleno-[5,6-*b*]quinolin-18-one **294** were obtained by reacting (*E*)-2-(arylidene)-3,4-dihydro-1(2*H*)-acridinone **290** with azomethine ylides generated from acenaphthoquinone **25** with sarcosine **20** or thioproline **22** in refluxing toluene, respectively (Figure 99) [139].

### 3.18. Thiazolidinones

A series of dispiro[indoline-3,2′-pyrrolidine-3′,5″-thiazolidines] of type **297**, which are potential *α*-amylase inhibitors (useful for type-2 diabetes mellitus), were obtained through the cycloaddition of azomethine ylide (generated from glycine methyl ester **295** and isatin **19**) and 5-arylidine-2-thioxothiazolidin-4-one **296** (Figure 100) [140].

Another group of benzo[*h*]quinolinyl dispiro-compounds **299**–**301** was obtained by reacting [5-(2′-chlorobenzo[*h*]quinolin-3′-yl)methylidene]-thiazolidin-2,4-dione/2-thioxothiazolidin-4-one **298** with various azomethine ylides formed from isatin **19** and different amino acids (sarcosine **20**, thioproline **22,** or *L*-proline **27**) (Figure 101) [141].

### 3.19. Thiazolo[3,2-a]pyrimidine-3-ones

Various (*E*)-arylmethylene-octahydro/decahydro cycloalka[*d*]thiazolo[3,2-*a*]pyrimidine-3-ones of type **302** reacted smoothly with azomethine ylides formed from isatin **19** and sarcosine **20** or thioproline **22** in a refluxing methanol-dioxane (1:1) mixture, thereby affording the corresponding spiro-oxindoles **303** and **304**, respectively (Figure 102) [142].

### 3.20. Benzo[1,4]thiazines

Spiro-oxindoles **306** and **307** and spiro-acenaphthylen-1-ones **308** and **309** were synthesized via a multicomponent reaction of 2-(4-methylbenzylidene)-4*H*-benzo[1,4]thiazin-3-one **305** and azomethine ylides derived from isatin **19** or acenaphthenequinone **25** with sarcosine **20** or *L*-proline **27** in refluxing toluene (Figure 103) [143].

## 4. Cyclic Unsaturated 2π-Electron Components

### 4.1. Non-Aromatic Cyclc 2π-Electron Components

#### 4.1.1. Alicyclic Unsaturated 2π-Electron Components

##### Intermolecular Cycloaddition Reactions

Cyclopentenone

The reaction of azomethine ylide generated from benzyl(methoxymethyl)(trimethylsilylmethyl)amine **1** with cyclopentenone **310** afforded bicyclic ketone **311** via an addition to the *C*2-*C*3 unsaturated linkage. Some analogs of **307** exhibited histamine H_3_ receptor antagonists that are responsible for the production and regulation of histamine and other neurotransmitters (Figure 104) [144].

1,4-Naphthoquinone

Spirooxindoles of type **313** were obtained by the cycloaddition of azomethine ylides, formed from isatin **19** and sarcosine **20**, with 1,4-naphthoquinone **312** in refluxing ethanol (Figure 105). Some of the synthesized compounds showed antibacterial activities against *Staphylococcus aureus*, *S. aureus* (MRSA), *Enterobacter aerogens*, *Micrococcus luteus*, *Proteus vulgaris*, *Klebsiella pneumonia*, *Salmonella typhimurium*, and *Salmonella paratyphi-B,* and antifungal activities against *Malassesia pachydermatis*, *Candida albicans,* and *Botyritis cinerea* relative to Streptomycin and Ketoconazole (standard references) [145].

Further spiro[benzo[*f*]isoindole-1,3′-indolines] of type **315** were synthesized by the cycloaddition of azomethine ylides (formed from isatin **19** and 2-(3-methyl-5-styrylisoxazol-4-ylamino)acetic acids **314**) and 1,4-naphthoquinone **312** using ceric ammonium nitrate (CAN) as a catalyst (Figure 106). Some of the products showed anti-inflammatory (determined via rat carrageen paw edema assay) and analgesic (determined via acetic acid writing protocol) properties relative to Ibuprofen and Diclofenac as references, respectively [146].

Tricyclic benzo[*f*]isoindole-4,9-dione-1-carboxylate **317** was obtained by reacting 1,4-naphthoquinone **312** with azomethine ylide generated from sarcosine ethyl ester hydrochloride **103** and paraformaldehyde **316** in the presence of iodine and sodium bicarbonate as a base in refluxing acetonitrile (Figure 107) [147].

##### Intramolecular Cycloaddition Reactions

The azatricyclic [6-5-7] ring system **320** was created via the intramolecular [3+2]-cycloaddition reaction of azomethine ylide generated from aldehyde **318** and *N*-(trimethylsilyl)methyl iminium salt **319** in the presence of a catalytic amount of phosphoric acid in DMF as a solvent (Figure 108) [148].

### 4.2. Aromatic Cyclic Unsaturated 2π-Electron Components

Azomethine ylide’s cycloaddition to aromatic 2π-electron components (aromatic or heteroaromatic) was reviewed in [149].

A series of benzoazepine-fused isoindolines of type **322** were obtained through thermal azomethine ylide-based cycloaddition of benzaldehydes bearing 3,5-dinitrophenyl **321** and N-substituted α-amino acids. The reaction was assumed to proceed through a regioselective dearomatizing [3+2] cycloaddition with the removal of HNO_2_, thus yielding the aromatic final product **322** (Figure 109) [150].

Nitro-substituted benzenes **323**–**329** underwent [3+2] cycloaddition of azomethine ylide derived from (*N*-(methoxymethyl)-*N*-(trimethylsilyl-methyl)-benzylamine) **1,** affording the pyrrolidinyl cycloadducts **330**–**337** (Figure 110) [151].

Pyrrolo[3,4-*c*]pyridines **339** were obtained through azomethine ylide’s (formed from sarcosine and paraformaldehyde **316**) cycloaddition with 3-nitropyridines **338** in refluxing toluene (Figure 111) [152].

Analogously, heterocyclic compounds bearing nitro groups **340**–**345** and **352**–**356** underwent a cycloaddition reaction with azomethine ylide derived from (*N*-(methoxymethyl)-*N*-(trimethylsilyl-methyl)-benzylamine) **1**, affording the pyrrolidinyl-containing analogs **346**–**351** and **357**–**361** (Figure 112 and Figure 113) [151,153].

Double cycloadducts of type **363** were obtained through azomethine ylide (formed from sarcosine and paraformaldehyde **316**) with meta-dinitro-containing nitrogenous heterocycles **362** in refluxing toluene (Figure 114) [154].

4-Chloro-5,7-dinitro-4-benzofurazan bearing indolyl heterocycle **364** underwent azomethine ylide (formed from the condensation of sarcosine and paraformaldehyde **316**) cycloaddition in refluxing benzene, affording the corresponding tetrahydro-5*aH*-[1,2,5]oxa-diazolo[3,4-*e*]isoindole **365**. Alternatively, conducting the reaction in refluxing MeCN afforded a mixture of **366** and **367**. Similarly, analogs with a pyrrolidinyl function (**366** and **367**) were obtained upon reacting the appropriate analog of **365** in MeCN at room temperature (in the darkness) (Figure 115) [155].

The cycloaddition reaction of azomethine ylide derived from (*N*-(methoxymethyl)-*N*-(trimethylsilyl-methyl)-benzylamine) with 4-nitrobenzofuroxan **368** afforded either mono **369** or bis **370** cycloadducts based via the substitution of the starting benzofuroxan at the 7-position (Figure 116) [156].

### 4.3. Heterocyclic Unsaturated 2π-Electron Components

#### 4.3.1. Maleimides

Spiro[3*H*-indole-3,2′(1′*H*)-pyrrolo[3,4-*c*]pyrroles] of type **372** were obtained in good yields by the cycloaddition of azomethine ylide (formed from sarcosine **20** and isatin **19**) to the *C*3-*C*4 unsaturated bond of maleimide **371**. Some of the synthesized compounds revealed promising to moderate antiproliferative properties against HEPG2 (liver), HCT116 (colon), and MCF7 (breast) cancer cell lines (SRB technique) relative to Doxorubicin (standard reference drug) (Figure 117) [157].

A microwave-assisted multi-component reaction of maleimide **371** with azomethine ylide produced from sarcosine **20** and ninhydrin **44** stereoselectively afforded spiro[indene-2,1′-pyrrolo[3,4-*c*]pyrroles] **373**. Some of the products showed promising antimycobacterial (*M. tuberculosis* H37Rv) properties relative to Cycloserine (Figure 118) [158].

In another reaction of *N*-phenylmaleimide **371** with azomethine ylides generated from 2-chloro-quinoline-3-carbaldehydes **374** and sarcosine **20**, two isomeric cycloadducts, namely, 1,4-diaza-bicyclo[3.3.0]octanes **375** and **376,** were formed (Figure 119) [159].

Tetracyclic pyrroloisoquinolines of type **378** were synthesized by the reaction of azomethine ylides, formed from isoquinolines **11** and phenacyl bromide **377**, with *N*-arylmaleimides **371** in the presence of cetyl trimethyl ammonium bromide (CTAB) (Figure 120) [160].

Bicyclic hexahydropyrrolo[3,4-*c*]pyrrole-1-carboxylates **380** and **381** were obtained by reacting *N*-phenylmaleimide **371** with a series of azomethine ylides generated in situ from sulfanyl-substituted imines of glycine esters **379** (Figure 121) [161]. Some of the synthesized diastereomeric compounds showed antioxidant activity relative to Nordihydroguaiaretic acid and Trolox [161].

Another reaction of maleimide **371** with pyrazole-4-carbaldehyde **382** and *α*-amino acid ester **383**, proceeding via azomethine intermediates in refluxing toluene, afforded the corresponding pyrazolylpyrrolopyrrole **384** (Figure 122) [162].

Isomeric pyrrolo[3,4-*a*]pyrrolizines **386** and **387** were synthesized by the cycloaddition of maleimide **371** with azomethine ylides formed from 3-alkylsulfanyl-2-arylazo-3-(pyrrolidin-1-yl)acrylonitriles of type **385** in refluxing benzene (Figure 123) [163].

#### 4.3.2. Maleic Anhydride

3,4-Dihydropyrrolo[2,1-*a*]isoquinoline **389** was obtained through the reaction of maleic anhydride **388** with azomethine ylide formed via the oxidation of tetrahydroisoquinoline **53** by dirhodium(II)caprolactamate [Rh_2_(cap)_4_] in the presence of *tert*-butyl hydroperoxide (TBHP) (Figure 124) [37].

#### 4.3.3. Benzo[b]thiophene-1,1-dioxide

The reaction of benzo[*b*]thiophene-1,1-dioxide **390** with a thermally generated azomethine ylide from aziridines **391** in refluxing dry benzene afforded the cycloadducts of type **392** (Figure 125) [164].

Three isomeric cycloadducts, **393**–**395**, were obtained by reacting benzo[*b*]thiophene-1,1-dioxide **390** with azomethine ylides generated from sarcosine **20** and aldehydes **13** in refluxing toluene (Figure 126) [164].

#### 4.3.4. Benzo[*c*]isoxazole and Benzo[*c*]isothiazole

Decahydroisoxazolo[3,4-*e*]pyrrolo[3,4-*g*]isoindole **397** and its isothiazolo-derivative **399** were synthesized by the [3+2]-cycloaddition of benzo[*c*]isoxazole **396** and benzo[*c*]isothiazole **398**, respectively, to azomethine ylide generated from sarcosine **20** and paraformaldehyde **316** in toluene under reflux conditions (Figure 127) [165].

#### 4.3.5. Indoles

Hexahydropyrrolo[3,4-*b*]indoles of type **402** were synthesized by reacting 3-nitroindoles of type **400** with azomethine ylides formed from *α*-amino acids (sarcosine **20** or *N*-benzylglycine **401**) and paraformaldehyde **316** (Figure 128) [166].

#### 4.3.6. Lactones

The reaction of *α*,*β*-unsaturated lactones of type **403** with azomethine ylide formed from *N*-methyl isatin **19** and proline **27** in refluxing toluene afforded the corresponding pyrrolidinyl-spirooxindole lactones of type **404** in high yield (Figure 129) [167].

Another glucosyl *α*,*ß*-unsaturated-7,3-lactone **405** reacted with azomethine ylides generated from isatin **19** and secondary amino acids (proline **27**, thioproline **22** or pipacolinic acid **142**) in refluxing dry toluene under N_2_ (inert atmosphere) to produce glucosylspiro-oxindoles **406**–**408** in a highly regio- and stereoselective manner (Figure 130) [168].

#### 4.3.7. Chromenes

3-Nitrochromenes of type **409** underwent a reaction with azomethine ylides formed from isatin **19** and amino acids (sarcosine **20**, proline **27,** or pipacolinic acid **142**) in refluxing toluene to afford the corresponding spiropyrrolidine/spiro-pyrrolizidine/spiroindolizidine-oxindoles **410** and **411** (Figure 131) [169].

Spiropyrrolidine-oxindole carbohydrate **413** was synthesized by the reaction of glycol 3-nitrochromene **412** with azomethine ylide formed from isatin **19** and sarcosine **20** in refluxing acetonitrile (Figure 132) [170].

Isomeric benzopyrano[3,4-*c*]pyrrolidines **415** and **416** were obtained via the cycloaddition of 3-nitro-2-trihalomethyl-2*H*-chromenes **414** to azomethine ylide generated from sarcosine **20** and paraformaldehyde **316** in refluxing toluene (Figure 133) [171].

However, the reaction of 2-aryl-3-nitrochromenes **409** with azomethine ylides formed from paraformaldehyde **316** and sarcosine **20** or *N*-benzyl-glycine **401** in toluene under refluxing conditions afforded the corresponding 3*a*-nitro-4-aryl benzopyrano[3,4-*c*]pyrrolidines of type **417** in high yields. Further, ^1^H,^1^H-NOE spectroscopic studies supported the structure of **417** (Figure 134) [172].

#### 4.3.8. Coumarins

A cycloaddition strategy for the synthesis of [1]-benzopyrano[3,4-*c*]pyrrolidines **419** and **420** was based on the reaction of 3-substituted coumarins of type **418** and in situ-generated azomethine ylides formed from sarcosine **20** or proline **27** with paraformaldehyde **316** in refluxing benzene (Figure 135) [173,174].

The reaction of 3-acetyl-2*H*-chromen-2-one **421** with azomethine ylides generated from isatin **19** and sarcosine **20** in refluxing toluene afforded the corresponding chromeno[3,4-*c*]spiropyrrolidine-oxindoles of type **422**, while the analogous reaction in methanol gave chromeno[3,4-*c*]spiropyrrolidine-oxindole derivatives of type **423** (Figure 136) [175].

Mixtures of isomeric benzopyrano[3,4-*c*]pyrrolidines **425** and **426** were obtained from the reaction of coumarin **418** with azomethine ylides formed from *α*-iminoester **424** in the presence of silver(I)-trifluoroacetate (AgTFA) in tetrahydrofuran at room temperature (Figure 137) [176].

#### 4.3.9. Chromones

Benzopyranopyrrolidine derivatives of type **428** were synthesized by the cycloaddition of 3-substituted chromones of type **427** with azomethine ylide generated from sarcosine **20** and paraformaldehyde **316** in benzene under refluxing conditions (Figure 138) [177].

Udry et al. described the synthesis of enantiomerically pure cycloadducts (**431a**, **431b**) from stabilized azomethine ylides of type **430** and sugar-derived enones (**429a** and **429b**) through the [3+2]-cycloaddition reaction in the presence of silver acetate (AgOAc) and DBU in acetonitrile. The cycloadducts were further used to synthesize enantiomeric polyhydroxyalkylpyrrolidines as potential *β*-galactofuranosidase inhibitors (Figure 139) [178].

#### 4.3.10. Isatoic Anhydride

1,3-Benzodiazepin-5-ones of type **433** were obtained through azomethine ylide (formed from (*N*-(methoxymethyl)-*N*-(trimethylsilyl-methyl)-benzylamine) **1** cycloaddition to isatoic anhydride **432** in trifluoroacetic acid in the presence of molecular sieves (4 Å) (Figure 140) [179].

## 5. Conclusions and Outlook

Among various methods, the [3+2]-cycloaddition reaction of azomethine ylides is one of the most adopted protocols for the formation of pyrrolidine and pyrrole systems. The chemistry of azomethine ylides has progressed significantly in the last two decades. Azomethine ylides have been used for the synthesis of many stereoselective natural products, core ring systems of natural products, and several bioactive molecules containing multiple chiral centers. The cycloaddition of a three-atom component to an appropriate unsaturated substrate, namely, the unsaturated 2π-electron component, is the most embraced approach to the synthesis of five-membered heterocyclic compounds. By using various unsaturated 2π-electron components in reaction with in situ-generated azomethine ylides, a plethora of pyrrolidinyl-containing heterocycles can be obtained in a highly regio- and stereoselective manner. As a result of intermolecular cycloadditions, one new ring with a defined stereochemistry is formed; however, when the three-atom component and the substrate are part of the same molecule, the cycloaddition is intramolecular and leads to a more complex molecular architecture that is difficult to access by other routes, namely, through the use of new bicyclic systems.

This review summarizes the synthesis of some of the most important compounds resulting from the [3+2]-cycloaddition reactions of azomethine ylides with various olefinic (acyclic, alicyclic/heterocyclic, and exocyclic) unsaturated 2π-electron components and highlights their potential therapeutic significance. We believe the compiled subject will develop interest within this field among the research community and encourage them to develop a wider variety of asymmetric [3+2]-cycloaddition reaction strategies for the synthesis of complex molecules.

## Data Availability

Not applicable.

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
