# Peer review of "Azomethine Ylides—Versatile Synthons for Pyrrolidinyl-Heterocyclic Compounds"

_molecules, 2023, doi:10.3390/molecules28020668_

Round 1

Reviewer 1 Report

This review article describes the synthesis of pyrrolidine derivatives from 1,3-dipolar cycloaddition reactions of azomethine ylides with various olefinic dipolarophiles. Authors briefly discussed about dipolar cycloadditions of azomethine ylides and synthesis of five-membered heterocyclic systems through azomethine ylides. The focus is on listing the literature examples and divided them into three sections based on the type of dipolarophiles (acyclic, exocyclic, cyclic) used to achieve the desired cycloaddition reactions. Authors presented the article very well by drawing the complex stereochemical structures of the spirocyclic compounds and included the biological activities of some compounds. It is a well written manuscript with adequate number of examples/references.

However, it is good idea to include a short paragraph to provide short summary of the published review articles / book chapter with similar focus (list/links provided below) and highlight the differences of current review articles from those published.

https://orgmedchemlett.springeropen.com/articles/10.1186/2191-2858-1-6

https://www.frontiersin.org/articles/10.3389/fchem.2019.00095/full

http://www.arkat-usa.org/get-file/44078/

The Chemistry of Heterocyclic Compounds, Volume 59: Synthetic Applications of 1,3-Dipolar Cycloaddition Chemistry Toward Heterocycles and Natural Products. Edited by Albert Padwa and William H. Pearson.

Additionally, I recommend the authors to check the following:

1.    Instead of using the phrase “pyrrolidinyl-containing” in the title, I recommend the following changes:

“Azomethine ylides - versatile synthons for pyrrolidinyl-heterocyclic compounds” OR

“Azomethine ylides - versatile synthons for pyrrolidine-containing heterocycles”

2.    * Page 2, line 45 : “medicated” should be “mediated”.

3.    * Page 2, line 48 : I prefer to place “alkenes” in the bracket …acyclic dipolarophiles (Alkenes)

4.    * Page 2, line 56/57 : Reframe the sentence. Instead of “…basic supported solid mediated catalyst”, “…supported solid-base catalyst”

5.    * Page 2, line 61 : Doesn’t seem appropriate. Remove or move the sentence to introduction.

6.    * Page 8, line 151 (related to scheme 13, structure 45) : Why “dispiro”, it is also spiro similar to other compounds!!

7.    * Page 2, line 170 : It’s more appropriate to say “a,b-unsaturated polarophiles” instead of “2-Ene-1-one polarophiles”

8.    * Page 30, line 461 : That should be “indanones”

Considering the importance of this review article, I recommend for publication in “Molecules” after the suggested revisions.

Considering the presentation and importance of the review topic, I recommend for publication in “Molecules” after the minor revision as suggested.

Author Response

Date:               December 27, 2022

Subject:           Revised Manuscript ID molecules-2118431 

Dear Reviewer,

Thank you for reviewing our manuscript. We appreciate your helpful and valuable suggestions. We have reviewed them carefully and believe we have made all appropriate changes to our manuscript.

Comments and Suggestions for Authors:

This review article describes the synthesis of pyrrolidine derivatives from 1,3-dipolar cycloaddition reactions of azomethine ylides with various olefinic dipolarophiles. Authors briefly discussed about dipolar cycloadditions of azomethine ylides and synthesis of five-membered heterocyclic systems through azomethine ylides. The focus is on listing the literature examples and divided them into three sections based on the type of dipolarophiles (acyclic, exocyclic, cyclic) used to achieve the desired cycloaddition reactions. Authors presented the article very well by drawing the complex stereochemical structures of the spirocyclic compounds and included the biological activities of some compounds. It is a well written manuscript with adequate number of examples/references.

Comment 1

However, it is good idea to include a short paragraph to provide short summary of the published review articles / book chapter with similar focus (list/links provided below) and highlight the differences of current review articles from those published.

https://orgmedchemlett.springeropen.com/articles/10.1186/2191-2858-1-6

https://www.frontiersin.org/articles/10.3389/fchem.2019.00095/full

http://www.arkat-usa.org/get-file/44078/

The Chemistry of Heterocyclic Compounds, Volume 59: Synthetic Applications of 1,3-Dipolar Cycloaddition Chemistry Toward Heterocycles and Natural Products. Edited by Albert Padwa and William H. Pearson.

Response

Thank you for your suggestion and as suggested we have now summarized the references with a brief statement.

Comment 2

Additionally, I recommend the authors to check the following:

Instead of using the phrase “pyrrolidinyl-containing” in the title, I recommend the following changes:

“Azomethine ylides - versatile synthons for pyrrolidinyl-heterocyclic compounds” OR

“Azomethine ylides - versatile synthons for pyrrolidine-containing heterocycles”

Answer

Thank you. We have now changed the title to “Azomethine ylides - versatile synthons for pyrrolidinyl-heterocyclic compounds”

Comment 3   

Page 2, line 45 : “medicated” should be “mediated”.

Answer

We have now corrected the typo mistake.

Comment 4

Page 2, line 48 : I prefer to place “alkenes” in the bracket …acyclic dipolarophiles (Alkenes)

Answer

We have now revised as suggested.

Comment 5

Page 2, line 56/57 : Reframe the sentence. Instead of “…basic supported solid mediated catalyst”, “…supported solid-base catalyst”

Answer

We have now changed the sentence as suggested.

Comment 6

Page 2, line 61 : Doesn’t seem appropriate. Remove or move the sentence to introduction.

Answer

We have now deleted the sentence.

Comment 7

Page 8, line 151 (related to scheme 13, structure 45) : Why “dispiro”, it is also spiro similar to other compounds!!

Answer

We have now updated it.

Comment 8

Page 2, line 170 : It’s more appropriate to say “a,b-unsaturated polarophiles” instead of “2-Ene-1-one polarophiles”

Answer

We have now updated it

Comment 9

Page 30, line 461 : That should be “indanones”

Answer

It has been now changed to “indanones and indanediones”.

Reviewer 2 Report

The main topic of this review is generally interesting for potential readers. The view on the cycloaddition reaction should be hovewer substantially changed, according to actual state of knowledge. The detailed comments with the respective references are listed below:

Many decades ago, it was thought that the compounds involved in [3+2] cycloaddition reactions were 1,3-dipoles in nature. It just looked nice on the diagrams illustrating their electronic structure. Today we know that the situation is much more complicated. It turns out that a large part of these compounds does not have a polar nature at all, but a diradical, pseudoradical or carbenoid nature. Therefore, the use of the term "1,3-dipole" is completely unjustified [Eur. J. Org. Chem. 267–282 (2019)]. So, term "1,3-dipole" should be replaced to the "three atom component" (TAC). Next, "dipolarophile" should be replaced to "unsaturated 2pi-electron component", and "1,3-dipolar cycloaddition" to "[3+2] cycloaddition".

Actual state of the knowledge completly undermine the "concerted", ("pericyclic") nature of the [3+2] cycloaddition. All known [3+2] cycloaddition are characterised by difficult type reorganisation of electron density, which can be illustrated using novel ELF techniques [Molecules 21, 1319 (2016)]. In any cases the transition state exhibit not a aromatic nature (many decades ago it was only some postulate, actually completly not supported ). Next, actually, interesting examples of stepwise  [3+2] cycloadditions with the participation of zwitterionic or biradical intermediates are known [Organics, 1, 49 (2020)]. So, therms "concerted" and/or "pericyclic" must be replaced to "one-step". Next, schemes with illustration of transition states should be corrected. There is no reason to believe a priori that within the transition states of any of the analyzed reactions, two new sigma bonds will be formed simultaneously. I suggest you just remove the transient images. Lastly, the mechanistic background of the considerations about mechanism of [3+2] cycloaddition should be corrected accordingly.

According to issues mentioned above, the scheme 1 must be considered only as historical, outdated view on [3+2] cycloaddition. It can be preserved in the text, but the respective commen with the actual point of view on [3+2] cycloaddition and respective sheme should be included.

According to issues mentioned above, azomethine ylides should be drawn as the authors have done in Scheme 3. Marking +/- charges should be avoided because it is not known whether a particular ylide is truly polar in nature.

Paragraph 2.2
Authors described only cases of functionalised azomethine ylides cycloaddition to nitroalkenes. At this time however, some examples of most simple azomethine ylide (N-methyl azomethine ylide) [3+2] cycloadditions to conjugated nitroalkenes are known. These examples must be described in the first part of this paragraph [Scientiae Radices, 1, 26 (2022); Chemistry of Heterocyclic Compounds, 53, 1161 (2017)].  Next, the first sentence ("Nitroalkenes are reactive dipolarophiles that are intensively used by various researchers in cycloaddition reactions.") must be supported with the respectve references.

In XXI century, the AM1 quantumchemical calculations are completly unvaluable for the interpretation of the reactin mechanisms. So, respective sentences should be removed from the text.

Author Response

Date:               December 27, 2022

Subject:           Revised Manuscript ID molecules-2118431 

Dear Reviewer,

Thank you for reviewing our manuscript. We appreciate your helpful and valuable suggestions. We have reviewed them carefully and believe we have made all appropriate changes to our manuscript.

Comments and Suggestions for Authors:

The main topic of this review is generally interesting for potential readers. The view on the cycloaddition reaction should be hovewer substantially changed, according to actual state of knowledge. The detailed comments with the respective references are listed below:

Comment 1

Many decades ago, it was thought that the compounds involved in [3+2] cycloaddition reactions were 1,3-dipoles in nature. It just looked nice on the diagrams illustrating their electronic structure. Today we know that the situation is much more complicated. It turns out that a large part of these compounds does not have a polar nature at all, but a diradical, pseudoradical or carbenoid nature. Therefore, the use of the term "1,3-dipole" is completely unjustified [Eur. J. Org. Chem. 267–282 (2019)]. So, term "1,3-dipole" should be replaced to the "three atom component" (TAC). Next, "dipolarophile" should be replaced to "unsaturated 2pi-electron component", and "1,3-dipolar cycloaddition" to "[3+2] cycloaddition".

Response: We have now carefully checked and updated the manuscript as suggested.

Comment 2

Actual state of the knowledge completly undermine the "concerted", ("pericyclic") nature of the [3+2] cycloaddition. All known [3+2] cycloaddition are characterised by difficult type reorganisation of electron density, which can be illustrated using novel ELF techniques [Molecules 21, 1319 (2016)]. In any cases the transition state exhibit not a aromatic nature (many decades ago it was only some postulate, actually completly not supported ). Next, actually, interesting examples of stepwise  [3+2] cycloadditions with the participation of zwitterionic or biradical intermediates are known [Organics, 1, 49 (2020)]. So, therms "concerted" and/or "pericyclic" must be replaced to "one-step". Next, schemes with illustration of transition states should be corrected. There is no reason to believe a priori that within the transition states of any of the analyzed reactions, two new sigma bonds will be formed simultaneously. I suggest you just remove the transient images. Lastly, the mechanistic background of the considerations about mechanism of [3+2] cycloaddition should be corrected accordingly.

According to issues mentioned above, the scheme 1 must be considered only as historical, outdated view on [3+2] cycloaddition. It can be preserved in the text, but the respective commen with the actual point of view on [3+2] cycloaddition and respective sheme should be included.

According to issues mentioned above, azomethine ylides should be drawn as the authors have done in Scheme 3. Marking +/- charges should be avoided because it is not known whether a particular ylide is truly polar in nature.

Response: Scheme 1 has been re-drawn/corrected according to the instructions. Other schemes and Figures have also been updated as per the suggestion.

Comment 3

Paragraph 2.2: Authors described only cases of functionalised azomethine ylides cycloaddition to nitroalkenes. At this time however, some examples of most simple azomethine ylide (N-methyl azomethine ylide) [3+2] cycloadditions to conjugated nitroalkenes are known. These examples must be described in the first part of this paragraph [Scientiae Radices, 1, 26 (2022); Chemistry of Heterocyclic Compounds, 53, 1161 (2017)].  Next, the first sentence ("Nitroalkenes are reactive dipolarophiles that are intensively used by various researchers in cycloaddition reactions.") must be supported with the respectve references.

Response: The suggested references are now included in the manuscript.

Comment 4

 In XXI century, the AM1 quantum chemical calculations are completly unvaluable for the interpretation of the react in mechanisms. So, respective sentences should be removed from the text.

Response: We have now removed the suggested portion from the manuscript.

Round 2

Reviewer 2 Report

Authors improved the manuscript substantially. Some, minor points require however small corrections. In particular:

Terms "1,3-dipole", "dipolarophile" should be removed from all part of the manuscript. In my previous review i explain in detail, why this word is full erronous.

The aromatic nature of transition state for [3+2] cycloaddition reactions was generally undermined.  In my previous review i explain in detail, why this word is full erronous.

Term "Three Atom Component" was introduced to the organic chemistry by Domingo [Eur. J. Org. Chem. 267–282 (2019)]. This work should be cited in the introduced.

Fig.2.
The captions should be changed: "Hystorical, Huisgen view on the [3+2] cycloaddition reaction"

Author Response

Date:               December 30, 2022

Subject:           Revised Manuscript ID molecules-2118431 

Dear Reviewer,

Thank you for reviewing our manuscript. We appreciate your helpful and valuable suggestions. We have reviewed them carefully and believe we have made all appropriate changes to our manuscript.

Comments and Suggestions for Authors:

Authors improved the manuscript substantially. Some, minor pointsrequire however small corrections. In particular:

Comment 1. Terms "1,3-dipole", "dipolarophile" should be removed from all part of the manuscript. In my previous review i explain in detail, why thisword is full erronous.

Response: We have now carefully checked and updated the manuscript as suggested.

Comment 2. The aromatic nature of transition state for [3+2] cycloadditionreactions was generally undermined. In my previous review iexplain in detail, why this word is full erronous.

Response: We do agree and updated the possible transition states as suggested.

Comment 3 Term "Three Atom Component" was introduced to the organicchemistry by Domingo [Eur. J. Org. Chem. 267–282 (2019)]. This work should be cited in the introduced.

Response: We have now cited the reference in the manuscript.

Comment 4. Fig.2.The captions should be changed: "Hystorical, Huisgen view on the[3+2] cycloaddition reaction"

Response: We have now updated the caption of the figure
